# Effects of Cancer Stem Cells in Triple-Negative Breast Cancer and Brain Metastasis: Challenges and Solutions

**DOI:** 10.3390/cancers12082122

**Published:** 2020-07-31

**Authors:** Kha-Liang Lee, Gao Chen, Tai-Yuan Chen, Yung-Che Kuo, Yu-Kai Su

**Affiliations:** 1Division of Neurosurgery, Department of Surgery, Taipei Medical University-Shuang Ho Hospital, New Taipei City 23561, Taiwan; leekhaliang@gmail.com (K.-L.L.); b101101104@tmu.edu.tw (G.C.); wetrypio@gmail.com (T.-Y.C.); 2Taipei Neuroscience Institute, Taipei Medical University, Taipei 11031, Taiwan; 3Taipei Medical University (TMU) Research Center for Cell Therapy and Regeneration Medicine, Taipei Medical University, Taipei 11031, Taiwan; s03271@tmu.edu.tw; 4Graduate Institute of Clinical Medicine, College of Medicine, Taipei Medical University, Taipei 11031, Taiwan; 5Department of Neurology, School of Medicine, College of Medicine, Taipei Medical University, Taipei 11031, Taiwan

**Keywords:** triple-negative breast cancer, brain metastasis, cancer stem cells

## Abstract

A higher propensity of developing brain metastasis exists in triple-negative breast cancer (TNBC). Upon comparing the metastatic patterns of all breast cancer subtypes, patients with TNBC exhibited increased risks of the brain being the initial metastatic site, early brain metastasis development, and shortest brain metastasis-related survival. Notably, the development of brain metastasis differs from that at other sites owing to the brain-unique microvasculature (blood brain barrier (BBB)) and intracerebral microenvironment. Studies of brain metastases from TNBC have revealed the poorest treatment response, mostly because of the relatively backward strategies to target vast disease heterogeneity and poor brain efficacy. Moreover, TNBC is highly associated with the existence of cancer stem cells (CSCs), which contribute to circulating cancer cell survival before BBB extravasation, evasion from immune surveillance, and plasticity in adaptation to the brain-specific microenvironment. We summarized recent literature regarding molecules and pathways and reviewed the effects of CSC biology during the formation of brain metastasis in TNBC. Along with the concept of individualized cancer therapy, certain strategies, namely the patient-derived xenograft model to overcome the lack of treatment-relevant TNBC classification and techniques in BBB disruption to enhance brain efficacy has been proposed in the hope of achieving treatment success.

## 1. Introduction

After lung cancer, breast cancer is the second most frequently diagnosed origin of brain metastases [1]. Globally, breast cancer is the primary cause of cancer-related death in women and affects two million women annually, with more than 600,000 cases of mortality related to recurrence and metastases [2,3]. During the course of metastatic breast cancer, the central nervous system (CNS) is often involved later, with the incidence ranging from 25% to 46% [4]. Because of the improvement of early detection imaging and medications used for adjuvant and systemic therapy that have low blood brain barrier (BBB) penetrance, the actual and reported incidence of brain metastases in breast cancer is likely to increase. Risk factors for brain metastasis include age <35 years, higher histological grade, higher number of non-CNS metastatic sites, and the duration from the date of primary diagnosis [5]. Breast graded prognostic assessment (GPA) and modified breast-GPA have been noted to accurately predict overall survival (OS) in patients with breast cancer with brain metastases (*p* < 0.001 for both scores) [6] and are therefore, generally used in a clinical setting. The prognosticators of OS include age, the extent of primary disease control, the presence of extracranial metastases or leptomeningeal disease, Karnofsky Performance Status (KPS), and the availability of systemic treatment options [7]. Notably, the propensity to develop brain metastasis in advanced-stage breast cancer varies based on cancer subtypes [4,8,9].

## 2. Triple-Negative Breast Cancer and Brain Metastasis

Triple negative breast cancer (TNBC) accounts for 15% to 20% of breast cancers [10]. The diagnosis of the triple-negative subtype is made by excluding the expression or amplification of three biomarkers (the estrogen receptor (ER), the progesterone receptor (PR), and the human epidermal growth factor receptor 2 (HER2) protein), which are the oncogenic drivers and targets for breast cancer treatment. The disease typically presents as histologically high-grade-infiltrating ductal carcinoma [11], which mostly affects in younger women (age <40 years) [12]. Unlike the cancer subtypes involving the hormone receptor or HER2 markers that govern the choice of target therapy, the main aim of systemic treatment is to disrupt cancer cell survival in the TNBC subtype through chemotherapy regimens involving anthracyclines, alkylates, taxanes, and/or platinum [13,14,15]. Studies have proposed various TNBC classifications based on the identification of the following: (1) genomic expression, (2) histopathology, and (3) copy number and mutational analysis, in the hope of developing treatment-relevant classifications as a guide to treatment efficacy [16,17,18,19]. However, current researches have produced mixed results with varying conclusions. To date, patients with TNBC have the poorest prognosis, with the median progression-free survival (PFS) ranging from 3 to 4 months after the failure of first-line therapy, disease recurrence in one-half of early-stage patients and up to 37% of 5 year mortality rate after initial surgery [20,21].

The incidence of brain metastasis in advanced-stage breast cancer varies based on subtypes, with 30% to 46% of brain metastasis cases occurring in the triple-negative subtype, approximately one-third of the cases in the HER2-enriched subtype, and 14% of the cases in the luminal subtype [4,8,9]. Although most brain metastases occur at the advanced stages of cancer progression, TNBC usually spreads to the brain rapidly at earlier stages [11,22,23]. A 15 year cohort study reviewed the metastatic behavior of all breast cancer subtypes and observed that bone was the most common site of metastases for all early-stage breast cancer subtypes, except TNBC. Patients with basal-like TNBC had a higher rate of brain (odds ratio (OR), 3.7; 95% confidence interval (CI), 2.1–6.5), lung (OR, 2.5; 95% CI, 1.6–3.8), and distant nodal metastases (OR, 2.8; 95% CI, 1.8–4.5) but a significantly lower rate of liver (OR, 0.5; 95% CI, 0.3–0.8) and bone metastases (OR, 0.4; 95% CI, 0.2–0.6) compared with patients with the luminal cancer subtype. A similar pattern was found for non-basal triple-negative tumors, but they were not associated with fewer liver metastases [24]. Although the basal subtype is typically responsible for the aggressive behavior of TNBC in patients [25], no statistically significant difference was noted between the basal and non-basal biological subtypes regarding survival with brain metastases [26].

The incidence of brain metastasis in TNBC (BM-TNBC) varies significantly based on the disease stage. For instance, the 5 year cumulative incidence of the brain being the initial site of metastasis is 3%, 5%, and 10% for I, II, and III disease stages, respectively [27]. A case series reported that more than a quarter of BM-TNBC patients had brain metastasis as the first recurrence site [28]. In addition, patients with TNBC had the shortest interval (22 months) from primary early breast cancer to brain metastasis development compared with the luminal (63.5 months) or HER2-enriched (30 months) subtypes [22]. Furthermore, patients with BM-TNBC had a shorter median survival after brain metastasis development compared with the other subtypes (5–7 months vs. 10–18 months, respectively) [23,28,29]. A series comprised 433 patients with TNBC with variable metastatic sites and observed that median survival following a diagnosis of brain metastasis was 7.3 months. A longer median survival from the time of first recurrent brain metastasis was noted compared with those of subsequent recurrent (17.3 vs. 6.3 months, *p* = 0.008). However, patients with first recurrent brain metastasis were associated with shorter OS compared with those without brain metastasis (17.3 vs. 22.1 months, *p* = 0.006) [28]. In addition, a recent multicenter analysis of 219 patients with breast cancer with brain metastases reported the subtype switching between primary disease and brain metastases. The prevalence of any receptor discordance was 36.3%, and the receptor-specific discordance was 16.7% in ER, 25.2% in PR, and 10.4% in HER2 expression [30]. Notably, the discordance in receptor expression could be related to varying prognosis [31,32]. Therefore, a confirmatory biopsy should always be considered.

## 3. Current Treatment Practices and Challenges in BM-TNBC

Regardless of differences in primary origin, current treatment practices for brain metastasis are similar and involve a combination of loco-regional treatment, such as surgical resection, radiosurgery, or radiotherapy, and systemic treatment. Generally, loco-regional treatments provide survival benefits and improve quality of life, especially when there is a reasonable systemic treatment option for extracranial disease [33].

### 3.1. Loco-Regional Treatment

Solitary brain metastases can be surgically removed to treat the symptomatic cerebral mass effect, achieve cytoreduction in tumor burden, and for histologic diagnostic purposes [34,35,36,37,38]. Therefore, surgeries are prioritized, especially in circumstances wherein the confirmation of pathology is needed. Noninvasive loco-regional therapies, such as stereotactic radiosurgery (SRS) or whole-brain radiotherapy (WBRT), are alternative or adjunctive to surgical resection in those who are at high risk of or non-amenable to total resection. SRS is preferred over WBRT to both resection cavities (adjuvant administration) and any other non-resected brain metastases, especially in low tumor volume cases, owing to superior local control and less cognitive decline [39,40]. For patients non-amenable to SRS (e.g., large tumor volume), a hippocampal-sparing WBRT is reasonable and offers better prognosis (4 months or greater) given its deleterious cognitive effects [41]. Even though the brain metastasis of breast cancer origin is moderately sensitive to radiotherapy, BM-TNBC has been observed to be radio-resistant. For example, SRS provides valid control of brain metastasis even with multiple lesions, and it remains useful for the metachronous development of new lesions of all subtypes, but TNBC is the exception [42]. Cho et al. reported that patients with TNBC had the shortest interval to retreatment with WBRT or SRS compared with patients with the luminal subtype (hazard ratio (HR) = 3.12, *p* < 0.001). Moreover, TNBC is associated with the poorest outcomes (median survival: 2.0 vs. 3.5—4.1 months; HR = 1.87; 95% CI, 1.25–2.80) after WBRT compared with other subtypes of breast cancer in the setting of brain metastases [43].

### 3.2. Systemic Therapy

Systemic therapy in BM-TNBC is reserved for extensive intracranial lesions or recurrent disease detected through imaging follow-up, whereas the choice of regimens is based on the active agents against primary tumor [33]. Retrospective analyses revealed that the survival of patients with breast cancer with brain metastases has improved, owing to advances in systemic therapy, which have led to the better control of both extracranial and intracranial diseases; however, this fact does not hold true for the TNBC subtype. Therefore, breast cancer subtype continued to be an important prognostic factor, putting patients with HER2-positive breast cancer at a lesser risk most likely because of the excellent systemic control and CNS penetration of current anti-HER2 therapies [44,45,46,47,48]. Unfortunately, unlike the HER2 subtype that has several regimen selections with proven CNS efficacy, the treatment choice of BM-TNBC is limited to monotherapy or combination therapy of cisplatin, etoposide, and/or high-dose methotrexate, with an undetermined treatment efficacy [49,50,51,52,53]. In addition, the development of systemic chemotherapy regimens in TNBC patients is challenging because of the limitation of disease classification that provides clinical considerations and the lack of proven oncogenic drivers that target vast disease heterogeneity [16,54]. In current treatment practices for TNBC primaries, there are over one-half of the patients receiving chemotherapy who do not achieve pathologic complete response (pCR) [55]. Thus, the selection of chemotherapy regimens to treat BM-TNBC remains largely debatable. Furthermore, a series comprised 167 patients with breast cancer with brain metastasis, including a subgroup analysis of 39 patients with BM-TNBC, and revealed that 18% of triple-negative primaries had a change in receptor status with brain metastases (5/39 gained ER/PR and 2/39 gained HER2) [31]. Nevertheless, the therapeutic implication based on the observation of subtype switching in brain metastases and primaries is largely to be determined.

### 3.3. Current Clinical Trials in BM-TNBC

To date, despite all these approaches, the prognosis for patients with BM-TNBC remains the poorest (median survival after CNS recurrence ranges from 2.9 to 4.9 months) [4,56,57,58]. Therefore, several effective treatment strategies are warranted for BM-TNBC. We previously summarized the investigation of several novel target regimens for TNBC treatment in a cell function-based manner [59]. However, a few of them have been evaluated for CNS efficacy. Herein, regimens for BM-TNBC that are being investigated with the inclusion of CNS related outcomes in active clinical trials are listed (Table 1), namely DNA repair agents, immune checkpoint kinase 1 inhibitors, anti-angiogenic agents, brain-penetrating peptide drug conjugate, and pan-PI3K inhibitor.

#### 3.3.1. Inhibitor of Poly-ADP-Ribose-Polymerase (PARPi)

TNBC was more likely to be diagnosed in patients with breast cancer susceptibility protein (*BRCA)1*-positive germline mutation, and conversely, approximately 20% of patients with TNBC have *BRCA1/2* mutation [60,61], with a strong association with poor survival outcomes as well as the presence of *TP53, KDR, PIK3CA, ATM,* and *AKT1* in TNBC [62]. PARP inhibition re-sensitizes cancer cells to DNA damage, the repair of which relies on a *BRCA* pathway-dependent homologous recombination mechanism [63]. In the OlympiAD study (open-label, randomized, phase 3 trial), oral olaparib monotherapy provided a significant benefit over standard chemotherapy (capecitabine, vinorelbine, or eribulin), with the median PFS being 2.8 months longer and the risk of disease progression or death was 42% lower (7.0 vs. 4.2; HR = 0.58). Notably, in a subgroup analysis of 121 metastatic TNBC, marked improvements were noticed in comparison with the hormone-receptor positive group [64,65]. A previously completed prospective, multi-center, phase II trial of irinotecan–iniparib designed for patients with progressive BM-TNBC (*n* = 37) determined that intracranial clinical benefit rate (CBR) was approximately 30%, with PFS and OS 2.1 and 8 months, respectively. Even though the trial assessed 4 patients with intracranial partial response with 2 harboring a *BRCA* mutation, the sample was too small to draw definitive conclusions and associate dysfunctional homologous recombination with treatment sensitivity (NCT01173497) [66]. Another PARPi agent, Veliparib (ABT-888), is a potent inhibitor of PARP, has excellent oral bioavailability, and improves CNS efficacy in *BRCA*-mutated intracranial TNBC xenograft tumor models [67,68]. Currently, a phase II clinical trial is being conducted that compares cisplatin plus veliparib with cisplatin monotherapy in BM-TNBC with or without *BRCA* mutation (NCT02595905).

#### 3.3.2. Immune Checkpoint Molecule Inhibitor

Immunologic escape describes the stage where malignant clones have acquired the capability to evade the adaptive immune system. Programmed cell death 1 (PD-1) is a transmembrane protein expressed on T cells, B cells, and NK cells; it binds to the PD-1 ligand (PD-L1) and then directly inhibits tumor cell apoptosis, thereby promoting peripheral T effector cell exhaustion and conversion of T effector cells to Treg cells [69]. In a phase III randomized trial (IMpassion 130 trial), 902 patients who had not received treatment for metastatic TNBC were randomly assigned to nab-paclitaxel with either atezolizumab or placebo [70]. Overall, only a modest but statistically significant difference in PFS (7.2 vs. 5.5 months; HR, 0.80; 95% CI, 0.69–0.92) was observed, favoring the incorporation of atezolizumab. Currently, additional strategies are being developed to improve the efficacy of brain metastases treatment, including combination immunotherapy with other systemic therapy or with radiation (brain metastasis: NCT02886585; BM-TNBC: NCT03483012, NCT04348747), as well as other approaches. Notably, an exploratory biomarker analysis determined higher levels of tumor infiltrating lymphocytes (TILs), and levels of CD8 T cells were noted in correlation with higher response rates [71].

#### 3.3.3. Anti-Angiogenic Agent

Angiogenesis is considered a crucial target for cancer therapy. However, to date, prospective studies have not revealed the effects of incorporating angiogenesis inhibitors on OS in women with TNBC. Among therapeutic agents in this class, bevacizumab has been the most researched. Unfortunately, data consistently revealed that although the incorporation of bevacizumab can improve PFS, there is virtually no effect on OS [72,73,74,75,76]. A current clinical trial that involves bevacizumab treatment for BM-TNBC is being conducted as a phase II trial (NCT04303988) in combination with an anti-PD-L1 monoclonal antibody, SHR-1316.

#### 3.3.4. Brain-Penetrating Peptide Drug Conjugate

Most recently, ANG1005, a novel taxane derivative consisting of three paclitaxel molecules covalently linked to Angiopep-2 that was designed to cross the CNS barriers and penetrate malignant cells through the low-density lipoprotein receptor-related protein 1 (LRP1) transport system, was found to exhibit therapeutic activity during a phase II trial on patients with breast cancer with leptomeningeal carcinomatosis (*n* = 24) and recurrent brain metastases (*n* = 60). The study revealed benefits (stable disease or better) in 77% (intracranial) and 86% (extracranial) of patients with breast cancer involving all subtypes with brain metastases. Although the BM-TNBC subgroup had only 13 patients, an intracranial CBR of 46% was achieved [77].

## 4. Understanding the Biology of Brain Metastases in Triple-Negative Breast Cancer

Cancer cells must invade through the extracellular matrix (ECM), intravasate blood vessels, survive the turbid flow of the vasculature, escape from immune surveillance, arrest in the capillary bed, extravasate, and finally colonize a distant metastatic site [78,79]. Patients with TNBC have a shorter interval to brain metastasis development and the shortest CNS-related survival among all breast cancer subtypes, suggesting the innate ability of TNBC tumor cells to adapt to the brain. The development of brain metastases occurs through the seeding of circulating tumor initiating cells (cancer stem cells (CSC)) into the brain microvasculature. In addition to the sanctuary site of the brain where the functional barriers resist the penetration of systemic medical therapies, tumor growth is aided by the unique intracerebral microenvironment [80]. Because of the genetic predisposition and cellular adaptation mechanisms, crosstalk between CSCs and brain-resident cells has rendered the molecular makeup of brain metastasis different from that of the primary tumors as well as from the metastases at other sites. In addition, we previously reported that TNBC tumors exhibit more traits of CSCs than the other breast cancer subtypes [59]. Herein, we discuss the mechanisms of BM-TNBC, namely BBB extravasation and brain-specific microenvironment interplay in relation to the CSC biology (Figure 1). Apart from the identification of drug targets defined by intracerebral tumors, identifying targets in CSCs and understanding the molecular mechanisms that support circulating CSCs to succeed in BBB extravasation, intracerebral seeding, and outgrowth in the brain are crucial in developing therapeutic strategies.

### 4.1. Effects of CSC Biology on BM-TNBC

CSCs are multipotent stem cells responsible for the long-term clonal maintenance and growth of most human neoplasms. The coexistence of both the cycling and quiescent states of CSCs when they mutate or interact with their microenvironment further increases their heterogeneity and plasticity in overcoming the environmental stressors [81]. Among solid tumors, breast CSCs (BCSCs) were the first to be identified through the analysis of the expression of CD44/CD24 and ALDH1 in relation to the distinct levels of differentiation in the breast cancer population [82,83]. Their expressions are associated with therapy resistance, local recurrence, and distant metastasis, which clinically contribute to the poor therapeutic response, disease-free survival (DFS), PFS, and OS [84,85]. TNBC is highly associated with the existence of CSCs. For example, a TNBC cell line with mesenchymal origin, MDA-MB-231, is mostly composed (>90%) of cells that display the CD44+/CD24-/low BCSC immuno-phenotype [86,87]. When correlating with a perspective of CSC hypothesis in patients with TNBC, two independent biological levels, namely cellular and immune, are identified to stratify the prognostic and possible therapeutic classification [19]. Enriched ALDH1-expressing cells are an independent prognostic factor that predicts poor prognosis in patients with TNBC [88,89,90]. In addition, Notch, Hedgehog, and Wnt signaling pathways that regulate the maintenance of a stem cell niche, invasion capacity, and apoptosis escape also contribute to the therapy resistance phenotype of BCSCs during systemic chemotherapy treatment [91,92,93]. Notably, the activation of Wnt/β-catenin signaling, which is associated with the microglial-promoted colonization of breast cancer cells in the brain tissue [94].

#### 4.1.1. Epithelial–Mesenchymal Transition in CSCs Breaking Away from the Primary Bulk Tumor

Evidence suggests that in various cancers, a subpopulation of circulating tumor cells (CTCs) has the CSC phenotype during the initial metastasis process [95,96,97,98] through the activation of epithelial–mesenchymal transition (EMT), which endows them with inherent metastatic potential [99]. The activation of EMT facilitates the invasion and migration of cancer cells from the primary tumor site to intravasation and translocation to distant metastasis sites [99], whereas the activation of mesenchymal epithelial transition (MET) after extravasation and cell seeding may facilitate colonization [100]. EMT transcriptional factors and the mesenchymal marker vimentin are expressed at higher levels in CD44+/CD24-/low BCSC-like cells than in more differentiated epithelial CD44-/CD24+ cells [101]. In addition, the changes in ECM dynamics may contribute to the disruption of asymmetric stem cell division, leading to CSC overexpansion [102]. Hyaluronic acid, which constitutes the ECM structures, interacts with the cell surface protein CD44, enhancing CSC properties by activating the stem cell marker NANOG [103]. CSCs in TNBC cell lines display highly invasive properties with elevated expression of proinvasive genes, such as those for interleukin (IL)-1, IL-6, IL-8, and urokinase plasminogen activator (uPA) [86]. Together with the stimulation of the nuclear factor kappa-light-chain enhancer of activated B cell (NF-κB) signaling, stromal uPA activates matrix metalloproteinase-2 (MMP-2) production and becomes responsive on the surface of tumor cells to cleave any component of the ECM, thereby allowing a break through the basement membrane and facilitating the EMT process [104,105].

#### 4.1.2. Autophagy in Circulating CSCs Undergoing Circulatory Arrest and Evading Immune Surveillance

In patients with cancer, approximately 1 × 10^6^ cancer cells per 1 g of tumor enter the circulation daily. However, only a fraction of these cells survive and reach a distant niche [106]. It has been reported that tumor cells take a longer time to extravasate the brain because of the extra boundaries contributed by the BBB [107]. In lung cancer, it takes approximately 48 h to extravasate the brain, whereas it takes only 6 h to extravasate the liver. On the other hand, breast cancer cells require 2 to 7 days to extravasate the brain. Consequently, CTCs have a longer survival time within the cerebral vasculature compared with that in other metastatic sites [107]. Studies have revealed that the mutation or inactivation of cell cycle-regulating genes and apoptosis-inducing genes enable CSCs to escape apoptosis [108]. In addition to DNA repair systems, autophagy is one of the processes that is strongly associated with CSC physiology in TNBC cells, including tumorigenesis, differentiation, plasticity, migration, or invasion, as well as pharmacological, viral, and immunoresistance [109,110,111]. Autophagy is a cellular degradation process that is essential for promoting cellular adaptation to stressors, cytokine mediation, chromosome stability, stem cell microenvironment, and cellular homeostatic mechanism [109,112,113,114,115]. It has been determined that in triple-negative BCSC, the knock down of autophagy-related genes, such as *ATG8/LC3* and *ATG12*, or pharmacological blockade of autophagy flux decreases the CD44+/CD24-/low CSC phenotype [116], probably through the IL6/STAT3/JAK2 pathways that contribute to the conversion of non-CSCs into CSCs [109]. The autophagy process can be induced upon the loss of integrin-mediated cell attachment to the surrounding ECM [117], and it further protects cells from detachment-induced cell death, termed anoikis [118]. Two key autophagy proteins, BECLIN1 and ATG4, are upregulated in mammospheres compared with adherent cells and are needed for their maintenance and expansion [119,120]. Upon EMT and autophagy induction, the process aids in the evasion from immune surveillance programs, such as T-cell-mediated lysis [121,122] and inhibit NK-mediated tumor cell killing by degrading granzyme B [123,124].

### 4.2. Extravasation from the BBB

The BBB is a complex functional boundary that tightly regulates the transmission of molecular and cellular substances into the brain. It is made up of the large hydrophilic molecules of endothelial cells, pericytes, astrocytic endfeets, and neuronal cells, which have evolved into protective insulation for neuronal signaling. Junctional complex proteins include claudins, occludins, and zona occluden (ZO) proteins, and junctional adhesion molecules connect endothelial cells with continuous tight junctions that limit the transcytosis rate. Two basement membranes (endothelial and astrocytic), namely embedded pericytes and astrocytic endfeets, play a crucial role in the maintenance and regulation of BBB permeability. Mechanistically, pericytes regulate BBB permeability through the expressions of transporters and the physical constriction of the blood vessels, whereas astrocytes cover most of the BBB surface area functioning to couple the endothelial cells and pericytes [125,126,127].

#### 4.2.1. Chemokines Ligand Receptor System on Circulating CSCs Migrating through the BBB

Therefore, to enter the brain parenchyma, cancer cells must pass through microcapillary walls by opening tight junctions [128]. These junctions can be destabilized by cancer cells through the expression of cytokines, chemokines, and inflammatory mediators [129,130]. However, it is not yet clear how circulating CSCs affect the function of the BBB. Current potential mechanisms of the extravasation strategy of tumor cells include docking, locking, trans-endothelial migration, and adhesion to the subendothelial matrix, which mimic the mechanisms used by immune cells [131]. The CXCL12/CXCR4/CXCR7 ligand receptor system is the vital axis that regulates neuro-glio-vascular interactions and vessel growth during human brain development. In a traumatic brain injury murine model, the CXCL12/CXCR4 axis was stimulated to promote neural stem cell migration through the stimulation of MMP-2 secretion [132,133]. Nevertheless, the CXCL12/CXCR4 axis promotes metastasis and invasion across a wide variety of solid tumors. For example, the CXCL12/CXCR4 autocrine-positive feedback mechanism controls the survival and proliferation of neural progenitor derived glioblastomas under hypoxic stress [134,135,136]. In breast cancer brain metastases, the CXCL12/CXCR4 axis takes part in the homing, motility, and progression of metastases to regulate the migration of CTCs through the BBB [105]. Notably, BCSCs have high CXCR4 expression and are defined as a pro-metastatic subpopulation, whereas the expression of CXCL12 defines regions with a higher risk of causing metastasis for breast cancer invasion [137,138].

#### 4.2.2. Interaction between Brain Microvascular Endothelial Cells and Circulating CSCs Mediates the Tight Junction Disruption and BBB Destabilization

Moreover, cancer cell-derived exosomes have been noted to have multiple roles in the events of metastasis, with a key role in pre-metastatic niche formation through vascular remodeling and modulation of cellular behaviors in the pre-metastatic site. Metastatic TNBC cells secrete miR-105-enriched exosomes to down-regulate ZO-1 protein expression in endothelial cells, disrupting the endothelial cell barrier and increasing vascular permeability, thereby facilitating invasion and migration [139]. Reciprocally, brain endothelial cells actively influence tumor cell extravasation and proliferation, which are mediated by bilateral cell surface receptors and adhesion molecules, such as integrins, selectins, chemokines, and the receptors of the immunoglobulin superfamily (ICAM-1, VCAM1) [140]. Activated αvβ3 integrin in circulating TNBC cells appeared to interact with platelets causing thrombus formation, which facilitates the arrest of tumor cells within the vasculature [141]. Furthermore, the activation of αvβ3-integrin assists in the intracerebral growth of TNBC cells through the continued upregulation of vascular endothelial growth factor (VEGF) [142]. An in vivo study injected brain-seeking TNBC cell lines into the murine carotid artery and demonstrated that the VEGF secreted from the cells destabilized the brain microvascular endothelial cells through activation of angiopoietin-2 (Ang2) expression, which further mediated BBB disruption (changes in the tight junction between ZO-1 and claudin-5). The interaction plays a critical role in the initial cascade of colonization of TNBC in the brain tissue, because the administration of neutralizing Ang-2 peptibody prevented changes to the BBB integrity and further inhibited the formation of TNBC cells colonization into the brain [143].

### 4.3. Intracerebral Tumor Microenvironment

#### 4.3.1. Dormant Period of CSCs Adaptation to the Brain Microenvironment

Evidence supports dynamic crosstalk between cancer cells and the brain microenvironment contributed by surrounding stromal cells, immune cells, and ECM, which are crucial for tumor growth after cell seeding in the brain [144]. Nonetheless, most CTCs are likely to die even after successfully arresting and extravasating the brain [145]. The adaptation period following dissemination is typically preceded by a period of dormancy in which cells are in a slow cell-cycling state; this is mediated by the expressions of stemness-associated transcription factors such as the SOX2, SOX9, and Wnt signaling pathways [146], which can last up to several decades [147]. Further exit from dormancy requires evasion from the immune surveillance mechanism and angiogenic switch to form micro-metastases [148]. Striking overlap exists between the behaviors of CSCs in tumors and the behavior of cancer cells in a dormant state, especially in the context of tumorigenesis after cell seeding; this suggests that the dormant extravasated tumor shows the CSC phenotype [149,150]. A study revealed that a combination of intracellular and extracellular signals within the tumor microenvironment, namely regulation of quiescence, alteration of angiogenic response, and modulation of immune surveillance govern the regulation of the dormant state [151].

#### 4.3.2. Interaction between Reactive Astrocytes and CSCs Evading Immune Surveillance through the Activation of STAT3 Pathway

Astrocytes have long been recognized as a key stromal component and immunomodulator of metastatic brain tumors which have tumor-killing or tumor-promoting effects, likely reflecting the fact that these cells exist as distinct subtypes with distinct functions [152,153,154]. Several astrocyte subtypes have been identified, termed A1 and A2 [155]. A1 astrocytes are regarded as proinflammatory astrocytes, which are induced by classically activated neuroinflammatory microglia through the secretion of IL-1a, TNF, and C1q. In the event of CNS injury and neurodegenerative disease, A1 astrocytes lose the ability to promote neuronal survival, outgrowth, synaptogenesis, and phagocytosis, thereby inducing the death of neurons and oligodendrocytes [156]. A2 astrocytes are thought to promote tissue repair through the production of neurotrophic factors. It is thought that most tumor-associated astrocytes are likely to be of the A2 subtype, which highly expresses the phosphorylated (activated) form of the signal transducer and activator of transcription-3 (p-STAT3) pathway [153]. Notably, STAT3 signaling in tumor cell is a key signaling mechanism activated by cytokines and growth factor receptors that establish an immunosuppressive microenvironment during the early stages of breast carcinogenesis to promote tumor growth and metastases [157]. In addition, STAT3 expression in the BCSC-like subset of the TNBC cell line serves as a chemo-resistant marker [158]. Mechanistically, these p-STAT3+ reactive astrocytes block the access of immune cells, such as CD8+ cytotoxic T cell, to cancer cells through the upregulation of immunosuppressive molecules, such as programmed cell death–1 ligand 1 (PD-L1), VEGF-A, lipocalin-2, and tissue inhibitor of metalloproteinases 1 (TIMP-1). Furthermore, the crosstalk between microglia and reactive astrocytes contributes to the establishment of an immunosuppressive microenvironment, thereby facilitating the metastasis of breast cancer to the brain. The p-STAT3+ reactive astrocytes exhibit increased expression of the CD74 ligand, MIF (macrophage migration inhibitory factor), and increased binding to CD74+ microglia, which upregulate the expression of midkine (neurite growth-promoting factor 2 (NEGF2), a downstream target of the NF–κB signaling pathway that promotes the development of brain metastases. All pieces of evidence suggest the significance of the STAT3 pathway during the interplay between the TNBC CSCs and astrocytes contributing to the microenvironment.

#### 4.3.3. Tumor Progression through the Activation of PI3K/Akt Signaling by Interaction between Reactive Astrocytes and CSCs

In addition, the phosphoinositide 3-kinase (PI3K)-Akt pathway has been implicated as a regulator of brain metastasis. PI3K signaling promotes various cellular processes that include proliferation, survival, metabolism, and angiogenesis in response to extracellular signals that activate receptor tyrosine kinases or G-protein coupled receptors. Notably, a study revealed that PI3K signaling in human breast cancer brain metastasis samples was not only active in the metastatic cells but also in the CNS microenvironment. Furthermore, in vitro and in vivo demonstrations revealed that treatment with buparlisib (BKM120), a pan-PI3K class 1 inhibitor with excellent BBB penetrance, reduced the macrophage or microglia-induced invasion and glial-assisted infiltration of breast cancer cells into the brain parenchyma [159]. The Wnt pathway acts in a compensatory manner when the TNBC cell line is challenged with buparlisib. Effectively, the dual PI3K and Wnt pathway inhibitor works in synergy to promote cell viability and enhance antitumor efficacy in TNBC cell lines [160]. In addition, results demonstrate that rational combinations of the buparlisib with MEK1/2 inhibitor in mice bearing intracranial TNBC tumors (SUM149, MDA-MB-231Br, MDA-MB-468, or MDA-MB-436) improved survival for intracranial SUM149 and MDA-MB-231Br, but not MDA-MB-468 or MDA-MB-436, suggesting therapeutic potential for the treatment of some BM-TNBC [161]. Currently, a phase II clinical trial of BKM120 plus capecitabine is being conducted in patients with BM-TNBC (NCT02000882, Table 1).

#### 4.3.4. Brain-Specific PTEN Suppression in Maintaining CSCs Plasticity during Tumorigenesis

Furthermore, TNBC cells manifest distinct gene expression patterns after metastasizing to different organs because the interplay with the site-specific extrinsic signals affects subsequent metastatic outgrowth. For example, mutations of phosphatase and tensin homolog (PTEN), a tumor suppressor gene and negative regulator of PI3K-Akt signaling, are rarely observed in the primary breast or lung cancers; however, a 21% increase in PTEN mutations was observed in brain metastatic tumors [162,163]. In a study with 111 tissue samples of breast cancer brain metastases, loss of PTEN protein expression was noted in 48.6% of all samples. Notably, it is significantly associated with the TNBC subtype (67.5%, *p* = 0.001) [164]. Previous studies have revealed that PTEN inhibition increased the expressions of the stemness and mesenchymal markers, OCT4 and SNAIL in TNBC cell lines [165], suggesting the role of PTEN suppression in CSC maintenance of brain metastasis. In addition, loss of PTEN expression in TNBC cells seemed to be the essential trait acquired from the brain-specific microenvironment, which led to an increased secretion of cytokine chemokine ligand 2 (CCL2, chemo-attractant during inflammation) and the recruitment of Iba1+ myeloid cells that enhance tumor cells proliferation and reduce apoptosis [163,166]. PTEN overexpression was evidenced to attenuate the invasiveness and migration of breast cancer cells as well as astrocyte activation [164]. Mechanistically, in the brain-specific microenvironment, reactive astrocytes suppress PTEN expression in TNBC cells with the intercellular transfer of PTEN-targeting microRNA (miR-19a), through exosomes secretion, whereas the blockade of astrocyte exosome secretion rescues PTEN loss and suppresses brain metastasis in vivo [163]. Notably, the PTEN level in TNBC cells is restored after the cells leave the brain microenvironment. All these studies suggest that the critical role of plasticity in PTEN suppression might contribute to the CSC phenotype in brain-adaptive TNBC cells, thereby indicating new opportunities for effective anti-metastatic therapies in BM-TNBC [165,167,168,169].

## 5. Patient Derived Xenograft Model: A Solution to the Diverse TNBC Heterogeneity

Because TNBC has heterogeneous morphology, mutational phenotype, and signaling profile, the response to the same agent could be different among tumors [16,54]. Molecular subtyping of TNBC tumors has identified six Lehmann subtypes, namely basal-like (BL1 and BL2), immunomodulatory (IM), mesenchymal (MES), mesenchymal stem-like (MSL), and luminal androgen receptor (LAR) subtypes [16]. The molecular subtype has later been refined from six (TNBCtype) into four (TNBCtype-4) tumor-specific subtypes (BL1, BL2, M, and LAR) [170]. Briefly, BL1 tumors are associated with cell cycle and DNA-damage response genes (highest of *TP53* mutations; high gain/amplifications of *MYC, CDK6*, or *CCNE1*; and deletions in *BRCA2, PTEN, MDM2*, and *RB1*). BL2 tumors are associated with survival and proliferative mediated receptor tyrosine kinase signaling activity and share a highly proliferative phenotype. Both BL1 and BL2 exhibit excellent response to taxane (antimitotic agents), and BL1 achieves higher pCR with neoadjuvant cisplatin. M tumors are associated with stem-related genes, such as sarcoma family kinase (SRC), phosphoinositide 3-kinase (PI3K), or mammalian target of rapamycin (mTOR), and low claudin expression. The LAR subtype displays a luminal pattern of gene expression despite ER negativity (e.g., high levels of *FOXA1, GATA3, SPDEF*, and *XBP1*), with elevated mRNA and protein levels of androgen receptor, overlapping in 82% of cases with luminal-A or luminal-B intrinsic subtypes [17]. Among 124 patients with site-specific metastasis data in the TNBC subtype (GSE12276, GSE2034, and GSE2603), stratification by TNBC subtype did not show any statistical differences in brain (*p* = 0.1238) and lung (*p* = 0.0776) metastasis (incidence of brain metastasis in LAR (3/11 = 27.3%), M (6/35 = 17.1%), BL1 (3/44 = 6.8%), and BL2 (2/34 = 5.9%) subtypes). Additionally, the M subtype displayed a significantly higher frequency of lung metastasis (46%) compared to all other subtypes (25%) (*p* = 0.0388). Furthermore, the incidence of bone metastasis was significantly higher for the LAR subtype (46%) as compared to all other subtypes 16% (*p* = 0.0456), consistent with the preference of hormonally-regulated cancers to metastasize to bone [170].

Notably, at present, none of the studies has reached the level of guiding systemic treatment efficacy for various TNBC tumors. For over one-half of patients receiving chemotherapy who failed to achieve pCR, predictors are required to be identified for selecting suitable chemotherapy candidates [55]. Nevertheless, given the vast heterogeneity from patient-to-patient and from primary-to-metastatic site in TNBC, there is almost always a discrepancy between preclinical or clinical trials and real-world practice. For example, cell lines including MDA-MB-231 represent the mesenchymal origin of TNBC and are the commonly used in vitro and in vivo models for cancer research. However, according to molecular subtype classification, both MES and MSC subtypes (per Lehman’s classification) only contribute to a small subset among all patients with TNBC. Moreover, cell-line xenografts can adapt to in vitro growth, losing the original properties of the host tumor.

Recent developments in the patient derived xenograft (PDX) model seem capable of producing samples that are authentic to the host tumor, which replicate tumor growth, tumor diversity, and tumor progression, including metastatic potential [171]. Furthermore, in patients with TNBC, several applications of PDX models have been made in terms of drug screening [172,173,174,175,176], drug resistance profiling [177], prognostic and functional gene assessment [178,179], and metastatic assessment [180,181]. In the future, PDX models may become the trend in personalized therapy and minimize the need for expensive and prolonged randomized controlled trials.

## 6. Techniques in Opening BBB or Blood-Tumor Barriers

The BBB or brain-tumor barrier (BTB) that limits the penetrance of the therapeutic compound is a huge obstacle during treatment of the sanctuary site of the brain in brain metastases. Upon tumor progression and intra-tumor neo-angiogenesis, the BBB is disrupted and transformed into BTB, which involves the displacement of astrocytes and pericytes, loss of astrocytic endfeet connections, and changes in tight junctions between endothelial cells [182]. Although BTB is more permeable than the BBB, it retains critical aspects of the BBB, including expression of active efflux transporters in endothelial cells and tumor cells [183]. Therefore, heterogeneous permeability observed in the BTB contributes to suboptimal drug accumulation in brain tumors. Several techniques for opening the BBB or BTB have been discussed, such as disruption or bypassing of the BBB or BTB.

### 6.1. Osmotic, Chemical, or Microbubbling Disruption of the BBB and BTB

The BBB and BTB can be opened using an osmotic agent, most commonly mannitol, to increase drug permeability [184]. Osmotic shrinkage of brain microvascular endothelial cells results in the transient opening of the tight junctions for up to 2–3 h [185,186]. However, intra-arterial mannitol injection is limited by the difficulty in target focusing. Although transient flow arrest has been proven to be safe and efficacious by previous studies, this procedure is threatened by the risk of stroke [187]. Bradykinin increases the level of nitric oxide and induces vasodilation, thereby enhancing vascular permeability. However, a phase II clinical trial compared the synthetic bradykinin analog RMP-7 plus carboplatin with carboplatin monotherapy for the treatment of recurrent malignant glioma and revealed that when administered intravenously, no increase in CNS concentration was noted [188]. Nevertheless, a combination of focused ultrasound and circulating microbubbles is a physical way to transiently disrupt the BBB, with the disruption lasting up to 24 h [189] and leading to the increased delivery of chemotherapy, immunotherapy, and gene therapy, which correlate with improvements in treatment efficiency, tumor progression, and OS [190,191]. Another research on a mouse model demonstrated an improvement in the delivery of anticancer agents and a significant increase in the median survival time in patients with tumors, including glioma and HER2+ breast cancer with brain metastasis [192].

### 6.2. Bypassing the BBB and BTB: Convection-Enhanced Delivery and Intrathecal or Intraventricular Injection

Drugs are delivered through the placement of a cannula to the target region into the interstitial space, bypassing the BBB or BTB, and this is called convection-enhanced delivery (CED). Lidar et al. delivered paclitaxel through CED in high-grade gliomas and noted that 11 of 15 patients with imaging responses had a median survival of 7.5 months [193]. Nevertheless, the future challenges associated with CED include cannula design, cannula placement, intra-tumoral penetration, infusate reflux, and delivery tracking. Intrathecal or intraventricular injection refers to the infusion of the drug to the lumbar subarachnoid space or ventricular system and then the diffusion into the brain. Notably, to deliver much higher doses to the leptomeningeal space, intrathecal chemotherapy can be used as the primary treatment modality for neoplastic meningitis. However, drug penetration is limited in intra-parenchymal brain tumors [194,195].

## 7. Conclusions

We summarized current clinical practices and trials on patients with BM-TNBC, whereas multiple mechanisms underlying CSC biology alongside brain metastatic tumorigenesis were reviewed systemically, namely EMT, autophagy, immune evasion, niches, dormancy, proliferation, and tumorigenesis. Critical challenges in treating BM-TNBC were pointed out that included high propensity of developing brain metastasis with the lack of CNS-active regimens and inconclusive treatment-relevant disease classification to guide treatment efficacy.

The principle of choosing the chemotherapy regimen for brain metastasis largely depends on the active agents against primaries. However, treatment for patients with BM-TNBC is limited with cytotoxic chemotherapy because of the lack of proven effective therapeutic strategies that target the TNBC heterogeneity. Even though several molecular classifications for TNBC have been sequentially proposed during the last decade, researches have produced mixed results with varying conclusions. Furthermore, since TNBC is not as frequent as receptor positive breast cancer and the number of patients with brain metastasis is again lower, it has hindered the attention of drug development research. Therefore, despite the fact that a wealth of actively conducting clinical trials of systemic regimens has been designed for TNBC treatment [59], only a few of them has entered the stage of evaluating CNS efficacy in brain metastatic setting.

In present study, we reviewed several preclinical evidences signifying the association of CSCs with tumor cell survival, immune evasion, and brain-adapting mechanisms in BM-TNBC. Thus, the blockage of CSC maintenance by targeting the governing pathways in BM-TNBC development, such as PTEN mutation/suppression, STAT3 and PI3K activation, seemed capable of suppressing the disease. In addition, because of the additional functional boundaries in the brain microvasculature, the CNS accessibility of regimen use should always be considered. Furthermore, the disease heterogeneity of TNBC from patient to patient, as well as from primary-to-metastasis sites has complicated the identification of systemic targets for treatment in patients with BM-TNBC. Therefore, the PDX models have shed light on individualized drug screening and facilitate the drug developmental process.

Knowing the fact that brain metastatic tumors in TNBC have a unique multifaceted nature that is different from that of the other cancer subtypes and the metastases at other sites, our work provides a platform demonstrating the effects of CSCs in BM-TNBC. In our opinion, the current development of a therapeutic strategy for BM-TNBC is still primitive. Herein, we propose exploring these approaches to the BM-TNBC treatment challenges, namely (1) CSC-targeted and tailored therapeutic strategies alongside the disease progression and recurrence, (2) regimen selection that has been tested individually for the treatment efficacy, (3) CNS efficacy enhancement by novel drug designation or drug delivery with BBB disrupting/bypassing technique, in the hope of paving the pathway for future treatment success.

## Figures and Tables

**Figure 1 cancers-12-02122-f001:**
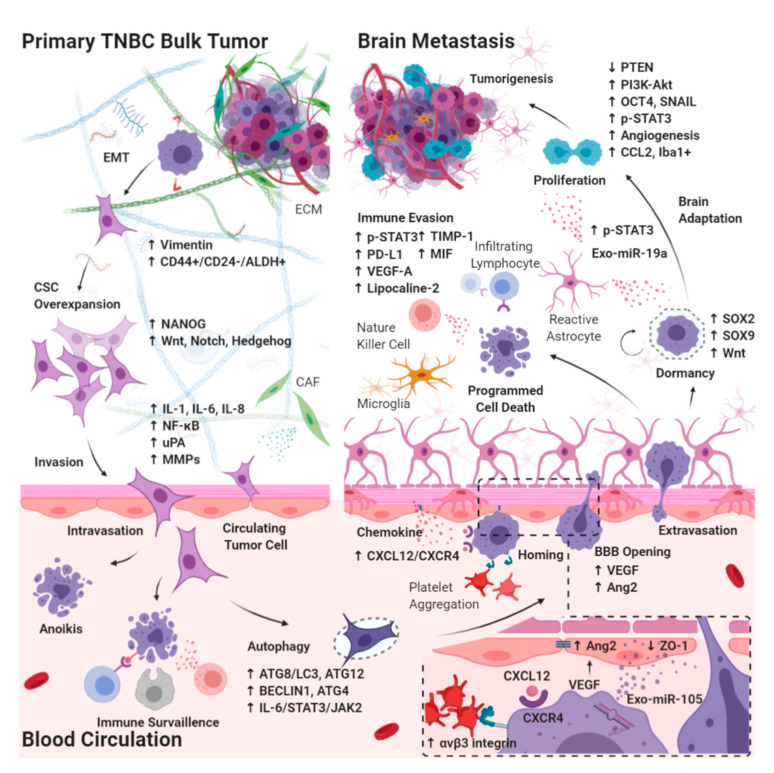
Diagram of molecular pathways involving CSC during the development of BM-TNBC. CSCs in TNBC bulk tumor must invade through the extracellular matrix in primary bulk tumors, intravasate blood vessels, survive the turbid flow of the blood circulation in vasculature, escape from immune surveillance, arrest in the capillary bed, and extravasate blood brain barrier (BBB), with subsequent tumor growth into brain metastasis involving microenvironmental niche–dormant cell interactions, neuroinflammatory cascades, and neovascularization. EMT: epithelial–mesenchymal transition; CAF: cancer-associated fibroblast; IL: interleukin; ATG: autophagy-related protein; VEGF: vascular endothelial growth factor; Exo-miR: microRNA enriched exosome; STAT: signal transducer and activator of transcription; JAK: janus kinase; uPA: urokinase plasminogen activator; MMP: matrix metalloproteinase; PD-L1: programmed death-1ligand 1; CXCL12: stromal cell-derived factor 1; CXCR4: C-X-C chemokine receptor type 4; Ang2: angiopoietin2; ZO: zonula occludens; SOX: SRY-box transcription factor; OCT4: octamer-binding transcription factor 4; PTEN: phosphatase and tensin homolog; TIMP: metallopeptidase inhibitor; MIF: macrophage migration inhibitory factor; SNAIL: zinc finger protein SNAI1; CCL2: monocyte chemoattractant protein-1; iba1+: ionized calcium binding adaptor myeloid cells.

**Table 1 cancers-12-02122-t001:** The clinical trials in triple-negative breast cancer with brain metastasis (BM-TNBC).

Target	Major Drug	Combinational Drug	Phase	NCT Identifiers	Status ^#^
BRCA1/2					
PARP	Iniparib	± Irinotecan	II	NCT01173497	C
	± Veliparib	Cisplatin	II	NCT02595905	Ac/NR
Angiogenesis					
VEGF	Bevacizumab	SHR-1316 + Cisplatin/Carboplatin	II	NCT04303988	Ac/NR
Immune checkpoint					
PD-1	Atezolizumab	SRS	II	NCT03483012	R
	Pembrolizumab	Anti-HER2/HER3 Dendritic Cell Vaccine Celecoxib Recombinant Interferon Alfa-2b	II	NCT04348747	Ac/NR
Brain-penetrating Peptide Drug Conjugate					
Taxane	ANG1005	-	II	NCT02048059	C
PI3K/AKT/mTOR Pathway					
PI3K	BKM120	Capecitabine	II	NCT02000882	C

^#^ C: completed; R: recruiting; Ac/NR: active, not recruiting.

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
