# Peer review of "Effects of Cancer Stem Cells in Triple-Negative Breast Cancer and Brain Metastasis: Challenges and Solutions"

_cancers, 2020, doi:10.3390/cancers12082122_

Round 1
Reviewer 1 Report
The review describes the state-of-the art around the formation of brain metastases in patients with TNBC and the treatments that are being implemented or trying to be implemented (clinical trials) with a wealth of numbers and percentages relating to the various types of patients. Authors also described some possible molecular basis (activation of STAT3, Pi3K-Akt path, and PTEN) to explain the different mechanisms of action deriving from the study of genomic expression and the use of this information for a better typing and treatment of patients in consideration of the high phenotypic heterogeneity of this cancer (Xenograft Model). They have also given much space to the description of local therapeutic treatments to overcome BBB and promote drug delivery.
The authors discussed in great detail much of what is known for TNBC and its propensity to metastasize in the brain but, even in the conclusions, there is no critical consideration of the described methodologies and they only suggest three lines of intervention, here too without comments or critical explanations.
A review, besides the historical description of the facts and state-of-the art of the problem treated, should also contain the critical opinion of the authors, which is missing.
Bibliography adequate.
Author Response
Q: The authors discussed in great detail much of what is known for TNBC and its propensity to metastasize in the brain but, even in the conclusions, there is no critical consideration of the described methodologies and they only suggest three lines of intervention, here too without comments or critical explanations. A review, besides the historical description of the facts and state-of-the art of the problem treated, should also contain the critical opinion of the authors, which is missing.
Answer: Thank you for the comments. We agreed on missing evidence on the three lines statement and the lack of critical opinions in the conclusion. Therefore, some modification has been made. Also, the treatment challenges part has been re-paragraph and explained in detail. Finally we proposed 3-line path for future therapeutic developement researches. (Please refer to line 532-566)
Reviewer 2 Report
In this manuscript, the authors aim at summarizing our current knowledge on TNBC brain metastasis in relation to cancer stem cells.
TNBC is the breast cancer subclass with the worst prognosis and metastasis in the brain contribute to this setback. Nevertheless, TNBC is not as frequent as receptor positive BCs and the number of patients with brain metastasis is again lower. However, I regard it as worthwhile writing a review on this topic. The review is packed with information but well structured and illustrated.
In the first part, the authors describe the epidemiology and outcome of TNBC related brain metastasis and current treatment options. Then current trials and actual developments such as PARP inhibition and immune-therapy are discussed.
The next part aims at understanding the biology of TNBC especially the development of brain metastasis. Here, the authors emphasize in particular the importance of cancer stem cells. For all steps starting from a tumour cell leaving the primary site, entering the blood stream and passing through the blood brain barrier, important molecular factors are discussed.
The next chapter discusses the advantages of PDX models especially in the context of tumour heterogeneity. This part on the TNBC subtypes, I had expected earlier in the manuscript. Also: Is there any information on the frequency of brain metastasis and the “Lehmann” subtypes available?
The 6th part deals with options to treat tumours “behind” the blood brain barrier.
Altogether from my point of view, the paper can be published when my question on the Lehman-TNBC subtypes has been answered sufficiently.
Author Response
Q: This part on the TNBC subtypes, I had expected earlier in the manuscript.
Answer: Thank you for the comments. We did initially put the TNBC classification much earlier in the introduction part. However, disproportionate description of TNBC and BM was noticed. Feedback from several readers suggested emphasizing the brain metastasis in the first few parts would be more comprehensive and focusing. Besides, we decided to combine the typing classification with the PDX part since the two topics are closely related.
Q: Also: Is there any information on the frequency of brain metastasis and the “Lehmann” subtypes available? Altogether from my point of view, the paper can be published when my question on the Lehman-TNBC subtypes has been answered sufficiently
Yes, there are actually reported prevalence of brain metastasis among different molecular subtype. Referring the published datasets with metastasis-site annotations (GSE12276, GSE2034 and GSE2603), Lehmann group had identified 124 patients with site-specific metastasis data and examined the metastatic pattern in TNBC subtype. Stratification by TNBCs subtype did not show any statistical differences in brain (p = 0.1238) and lung (p = 0.0776) metastasis [incidence of brain metastasis in LAR subtype (3/11 = 27.3%), M subtype (6/35 = 17.1%), BL1 (3/44 = 6.8%) and in BL2 (2/34 = 5.9%) subtype]. The M subtype displayed a significantly higher frequency of lung metastasis (46%) compared to all other subtypes (25%) (p = 0.0388). Metastasis to the bone was significantly different among TNBC subtypes (p = 0.0398). For example, the incidence of bone metastasis was significantly higher for the LAR subtype (46%) as compared to all other subtypes 16% (p = 0.0456), consistent with the preference of hormonally-regulated cancers to metastasize to bone. We thank for the comment and the prevelance part has been intergrated into the manuscript. (Please refer to line 468-475)
Reviewer 3 Report
This review article focuses on the subject of brain metastasis by triple-negative breast cancers (TNBCs). The review presents sufficient details on the TNBC subtypes, their propensities for brain metastases (BM-TNBC), and current treatment practices. In addition to discussing current clinical trials for BM-TNBCs, authors present a detailed description of biology of BM-TNBCs, various pathways including roles of cancer stem cells (CSCs), patient-derived xenograft models, and methods to penetrate blood-brain barrier for delivery of therapeutics to treat BM-TNBCs. The review is generally well structured, well written, and comprehensive on the subject of TNBC-associated brain metastasis. My minor suggestion for revision is noted below:
- Please move details in lines 239-248 to legend section of figure 1.
Author Response
This review article focuses on the subject of brain metastasis by triple-negative breast cancers (TNBCs). The review presents sufficient details on the TNBC subtypes, their propensities for brain metastases (BM-TNBC), and current treatment practices. In addition to discussing current clinical trials for BM-TNBCs, authors present a detailed description of biology of BM-TNBCs, various pathways including roles of cancer stem cells (CSCs), patient-derived xenograft models, and methods to penetrate blood-brain barrier for delivery of therapeutics to treat BM-TNBCs. The review is generally well structured, well written, and comprehensive on the subject of TNBC-associated brain metastasis. My minor suggestion for revision is noted below:
Please move details in lines 239-248 to legend section of figure 1.
Answer: Thank you for the comment. We have reformatted the paragraph into figure legend sections. (Please refer to line 231-245)